# Towards Fake News Detection: A Multivocal Literature Review of Credibility Factors in Online News Stories and Analysis Using Analytical Hierarchical Process

Muhammad Faisal Abrar [1], Muhammad Sohail Khan [2], Inayat Khan [3], Mohammed ElAffendi [4] and Sadique Ahmad [4,*]

1. Department of Computer Software Engineering, University of Engineering and Technology, Peshawar 25120, Pakistan; abrarfaisal49@gmail.com
2. Department of Computer Software Engineering, University of Engineering and Technology, Mardan 23200, Pakistan; sohail.khan@uetmardan.edu.pk
3. Department of Computer Science, University of Engineering and Technology, Mardan 23200, Pakistan; inayatkhan@uetmardan.edu.pk
4. EIAS Data Science and Blockchain Lab, College of Computer and Information Sciences, Prince Sultan University, Riyadh 11586, Saudi Arabia; affendi@psu.edu.sa
* Correspondence: ahmad01.shah@ieee.org

**Abstract:** Information and communication technologies have grown globally in the past two decades, expanding the reach of news networks. However, the credibility of the information is now in question. Credibility refers to a person's belief in the truth of a subject, and online readers consider various factors to determine whether a source is trustworthy. Credibility significantly impacts public behaviour, and less credible news spreads faster due to people's interest in emotions like fear and disgust. This can have negative consequences for individuals and economies. To determine the credibility factors in digital news stories, a Multivocal Literature Review (MLR) was conducted to identify relevant studies in both white and grey literature. A total of 161 primary studies were identified from published (white) literature and 61 were identified from unpublished (grey) literature. As a result, 14 credibility factors were identified, including "number of views", "reporter reputations", "source information", and "impartiality". These factors were then analysed using statistical tests and the Analytic Hierarchy Process (AHP) for decision-making to determine their criticality and importance in different domains.

**Keywords:** comprehensive synthesis of literature; information trustworthiness; success factors; motivators; digital news stories

## 1. Introduction

The preservation of information on the World Wide Web (WWW) is known as digital preservation [1]. This is important because the rapid changes in technologies and the short lifespan of digital objects make it challenging to access and preserve information on the WWW for future generations [2,3]. Digital preservation involves various activities, such as preserving the integrity and authenticity of digital objects, ensuring the long-term accessibility of digital information, and managing the risks associated with digital data [4–6]. A critical aspect of digital preservation is the credibility of the information being preserved [7]. Credibility refers to the trustworthiness and reliability of the information. It is essential to preserve credible information for future generations, as less credible information can have negative consequences, such as disrupting responses to terrorist attacks or natural disasters or affecting stock prices and large-scale investments [8,9]. Several factors can impact the credibility of the information being preserved on the WWW. These include the source of the information, the accuracy and thoroughness of the information, the clarity of the information's presentation, and the source's potential biases [10]. It is essential to consider

these factors when evaluating the credibility of the information being preserved [11]. The source of information is a crucial factor to consider when assessing the credibility of digital information. This includes the credibility of the news outlet or publisher and the credibility of the individual or organization providing the information [12]. News outlets and publishers with a history of accuracy and credibility are more likely to present reliable information. It is also essential to consider the credibility of the individual or organization providing the information. The information is more likely to be reliable if the source has a track record of credibility and expertise in the relevant topic. Besides considering the information source, it is also imperative to evaluate the accuracy and thoroughness of the information being preserved [13]. It includes ensuring that the information is supported by evidence and sources and that it is complete and up to date. It is also essential to consider any potential biases in the information, as these can impact its credibility [14]. The clarity of the presentation of the information is another factor to consider when evaluating the credibility of digital information. If the information is clear and understandable, it is more likely to be credible. This includes using appropriate language for the intended audience and ensuring that the information is organized logically and coherently [12,13]. Finally, it is vital to consider the potential biases of the source when evaluating the credibility of digital information. Bias can take many forms, such as political, financial, or personal bias. It is essential to be aware of any potential biases that may impact the credibility of the information being preserved [13,14]. Overall, the credibility of the information being preserved on the WWW is crucial to digital preservation. By considering the source, accuracy and thoroughness, clarity of presentation, and potential biases of the information, it is possible to ensure that only credible information is preserved for future generations [11–14]. Ensuring the reliability and trustworthiness of information accessed by users is crucial in order to avoid negative consequences. This is particularly important for news information, which is often generated from multiple sources in real-time. Thus, it is essential to assess the credibility of news articles based on various criteria, which should be developed systematically. The research aims to identify the factors affecting the credibility of digital news stories, and to address the lack of credibility, it is necessary to distinguish credible information from less credible information through a thorough examination of information diffusion. The study will conduct a multivocal literature review to identify credibility factors for digital news stories. This study is based on the following research questions.

1. What factors are identified in the literature (as well as in the grey literature) to be considered for ensuring the digital information credibility of digital news stories?
2. How should we rank the identified credibility factors using the Multi-Criteria-Decision-Making (MCDM) algorithm?

This paper is organized into several sections. Section 1 introduces the research topic and objectives. Section 2 reviews existing literature on digital news credibility. Section 3 presents the motivation and novelty. Section 4 explains the methodology, including the research goals, questions, and search strategy. Section 5 presents the results, divided into statistical tests and an analytical hierarchical process. Section 6 discusses implementation challenges and practical insights. Section 7 summarizes the main findings and contributions. Section 8 addresses the limitations and future directions.

## 2. Literature Review

The increasing digitization of society has made the Internet a central part of many people's lives. In addition to its uses for communication and entertainment, the Internet is an important tool for finding information. The amount of digital information available is continually growing, with more being added daily [15–20]. The Internet has become an essential resource for accessing a vast amount of information on a wide range of topics. It allows users to easily search for and find information on almost any subject they are interested in. This has made it an invaluable resource for students and professionals who need to access reliable and up-to-date information for their work or studies. However, the vast amount of online information can also be overwhelming, and it can be challenging

to determine the credibility and reliability of the sources. It is important for users to be discerning and to carefully evaluate the information they find online to ensure it is accurate and reliable. Overall, the Internet is essential for accessing and sharing information, but it is important to use it responsibly and to critically evaluate the information found online. The individuals involved in education, library services, and app development have a shared interest in disseminating information widely, without distinguishing between credible and less credible sources [21]. While the quality of information is crucial, the focus on promoting credibility is not prioritized as much since it is deemed the responsibility of the end-user [22]. With the prevalence of false information on online platforms, identifying trustworthy sources has become a challenging computational task. Scholars are now shifting their focus towards the nature of information rather than just information itself. This new approach is greatly contributing to research on credible information. Digital information platforms, such as social media, play a significant role in the dissemination of less credible information. These platforms provide information providers with a peer-review-less environment, as highlighted in [23]. However, our focus is on accessing information that plays a minor role in spreading inaccurate information. News, in particular, adds an interesting dimension to this issue. Whether it is political, social, sports, entertainment, or satirical news, it is a major source of interest for people. We suggest that this dimension could be either less credible or more credible. The spreading of news does not see its origin but keeps forwarding it to people, just going through the content at an abstract level. One highlighted reason for spreading news is reading it at a conceptual level. Typically, people do not understand the context and keep forwarding. If we specify the broad terminology of information, it will be news. The term news is considered fascinating in the reader's eye. In order to address the issue of credibility in news stories, it is necessary to differentiate between more and less credible information. To achieve this, it is important to examine the diffusion of information in-depth after distinguishing between credible and less-credible sources. The goal of this study is to identify these factors and their practices. While the increased number of websites has led to a greater volume and variety of available content, the credibility of such content remains a significant question in many situations. The assessment of credibility can be inaccurate if content groups, such as articles, discussion sources, and help, are not properly established [24]. With the rise of Web 2.0 and social media, the internet has become the primary source of information dissemination, with numerous news stories, audio, and videos of varying degrees of credibility. As a result, researchers have shifted their focus to the credibility of content. The terms "credibility" and "trust" are often used interchangeably, with "believability" also being considered an aspect of credibility. People assess credibility by considering multiple dimensions simultaneously, as highlighted by previous research [25–28]. The need for powerful credibility assessment abilities among users is emphasized, as they are the arbiters of accuracy in a domain where they may be informed or uninformed. Assessing information on the web requires awareness of potential bias and vested interests of content writers to determine whether information is appropriate, correct, or plausible. The digital information viewer faces a significant challenge in enhancing the credibility of content, as demonstrated by research from the Stanford University Persuasive Technology (STUP) Lab [9,14,29].

Fog B [29]: Fog B's article delves into the impact of various links and domains on people's perception of credibility and its implications. The research studies discussed in the article focus on the credibility of business and e-commerce websites. The authors in this study emphasize the importance of credibility assessments for web sites, particularly in critical areas such as finance and health. The study aims to identify gaps in consumer education and design guidelines for improving understanding of online credibility. Similarly, when it comes to digital news stories, consumers need to be able to assess the credibility of sources and evaluate the accuracy of the information provided. To accomplish this, they may need to improve their media literacy skills and develop critical thinking strategies. Overall, the study indirectly underscores the importance of credibility assessments not just for websites, but for digital news stories as well.

Maloney, Richard F and Beltramini [23,30,31]: These studies were reported from the literature on the credibility of advertisements in marketing. These research articles analyse the contents and a few attributes of promotion. One of our research's major primary goals is to identify the factors that ensure credibility. In the literature, we found that the perception regarding the advertisement's credibility includes the company's reputation and experience. The credibility of the information is measured via certain factors, i.e., the message contents, source credibility, source bias, and source reputation.

Austin et al. [32]: The study describes a research study that examines the effects of the message type and source reputation on judgments of news believability. The study uses a between-groups 3 × 3 factorial experiment with a total of 516 participants. The judgments are conceptualized as source credibility and assessments of apparent reality. The study finds that a more innocuous message results in more positive judgments of believability, but the reputation of the source has no direct effect on believability judgments, nor does it interact with the message type. The study concludes that some people base their judgments of news believability more on assessments of the apparent reality of the message content than on the reputation of the media source. Three indices combining measures of source credibility and message apparent reality emerge from a factor analysis, comprising judgments of source truthfulness and message accuracy, source expertise and message representativeness, and source bias and personal perspective.

Wathen et al. [33]: The study discusses how people decide what to believe when seeking information online. It highlights the multidimensionality of the concept of credibility, which is influenced by factors related to source, message, and receiver. However, there is a relative lack of high-quality research evidence, leaving key questions unanswered. To advance our understanding of these issues, research should focus on identifying key markers for credibility of online information, exploring the importance of surface characteristics of web-based media, identifying the best analogy for publication authority/credibility to web sites, and examining the accuracy of the proposed model for judging the credibility of web sites and the information presented on them. From a practical perspective, the text recommends designing an "ideal" web site that emphasizes a good interface, professional image, and source credibility. It also suggests taking advantage of interactive properties of the medium and tailoring content to the beliefs and needs of the audience. The study notes that the interactivity of computer applications is constantly evolving, and the use of human-like virtual agents may change the way users seek and use online information in the future. Credibility issues surrounding medical information have gained significant attention, as nearly 50% of internet users seek health-related information online. Given the importance of such information, many researchers have emphasized the need to assess the credibility of health-related information available on the web. In the case of news, credibility can be assessed by identifying less-credible stories, which can be defined as specific articles, including editorials, news reports, exposes, and other intentionally misleading content.

Rubin et al. [34]: This study discusses the importance of distinguishing between truthful and deceptive news reports, and presents a research project aimed at developing an automated approach for deception detection in news verification. The researchers analysed the rhetorical structures and coherence relation patterns of fabricated and authentic news reports, and used a vector space model to cluster the news by discourse feature similarity. The predictive model achieved 63% accuracy on the test set, which is comparable to human lie detection abilities but not significantly better than chance. The researchers note several confounding issues and methodological limitations that require further research, but suggest that a news verification system could improve credibility assessment of digital news sources and increase new media literacy to prevent the spread of fake news.

Metzger et al. [35]: The article discusses the challenges of locating trustworthy information in digitally networked communication environments and the increasing reliance on information available solely or primarily online. The article focuses on the use of cognitive heuristics in credibility evaluation and presents research findings that illustrate the types

of heuristics people use when evaluating the credibility and accuracy of online information. The article concludes with a call for further research to better understand the role and influence of cognitive heuristics in credibility evaluation in computer-mediated communication contexts. In summary, the article highlights the need to better understand how people make judgments about the credibility of online information, given the increasing reliance on digital media for information consumption.

Parth Patwa [36]: The paper introduces a dataset called "Fighting an Infodemic: COVID-19 Fake News Dataset" that addresses the problem of fake news and rumours related to COVID-19 on social media. The dataset consists of 10,700 manually annotated social media posts and articles, both real and fake, on COVID-19. The authors benchmark the dataset using machine learning algorithms and achieve a high performance of 93.46% F1-score with Support Vector Machine (SVM). The paper discusses related work in fake news detection, describes the dataset development process, and highlights the challenges associated with identifying and combating fake news. The dataset statistics reveal differences between real and fake news, such as the length of posts. The paper concludes by emphasizing the importance of tackling fake news during the COVID-19 pandemic and provides the dataset and code for further research.

Bilal Al-Ahmad [37]: The paper addresses the issue of misinformation and fake news related to COVID-19 during the pandemic. The authors propose an evolutionary fake news detection method using four models, aiming to reduce symmetrical features and achieve high accuracy. They apply three wrapper feature selection techniques and evaluate the performance on the Koirala dataset and six derived datasets. The proposed model achieves the best accuracy of 75.43% and outperforms traditional classifiers. The authors suggest applying the methodology to other domains and larger datasets for future work.

M Zivkovic [38]: The paper proposes the use of a modified ant lion optimizer (ALO) to address the issue of false news and disinformation during the COVID-19 epidemic. The ALO algorithm, inspired by the trapping technique of ant lions, is applied for feature selection and dimensionality reduction to enhance classification accuracy. Experimental results demonstrate that the proposed ALO-based technique outperforms other modern classifiers in terms of accuracy, providing an effective approach to combat false news related to COVID-19.

William Scott Paka [39]: The paper introduces the task of COVID-19 fake news detection on Twitter and presents the Cross-SEAN model. They collect a labelled dataset of genuine and fake COVID-19-related tweets, along with unlabelled data. The model incorporates tweet text, features, user information, and external knowledge from credible sources. Cross-SEAN outperforms seven state-of-the-art models and is implemented as a Chrome extension, Chrome-SEAN, which flags fake tweets in real-time. Limitations include potential noise in external knowledge and the need for improved robustness and early detection capabilities.

## 3. Motivation and Novelty

Ensuring the credibility of information is critical, especially in the digital age, where misinformation and fake news are prevalent. Consumers must be able to distinguish between fact and fiction to make informed decisions and opinions. Credible digital news stories can play a significant role in shaping public opinion, which in turn affects political, social, and economic systems. Thus, it is essential to recognize the role that credibility plays in the consumption of digital news stories and take steps to ensure that the information being consumed is accurate and trustworthy. This can promote informed and critical thinking, which ultimately leads to a more engaged and informed society. The credibility of digital news stories can vary, like any other form of information. Thus, it is crucial to evaluate the reliability of news articles carefully to ensure that the information being consumed is accurate and trustworthy. Several factors can impact the credibility of digital news stories, such as the reputation of the news source, the quality of journalism, and the evidence presented to support the claims made in the article. In addition, the presence of

errors or biased language in the article can also diminish its credibility. To fully address the issue of credibility in digital news stories, it is necessary to adopt a systematic approach to evaluate the reliability of the information. This could involve using a set of established criteria to assess the credibility of news articles or developing new criteria specifically tailored to the digital media landscape. It may also be necessary to consider the specific context in which the news is being consumed, as different audiences may have different standards for what they consider to be credible information. Overall, it is important to recognize the role that credibility plays in the consumption of digital news stories and to take steps to ensure that the information being consumed is accurate and trustworthy. This can help promote informed and critical thinking and lead to a more informed and engaged society.

## 4. Methodology

Multivocal Literature Reviews (MLRs) are a method used to summarize and evaluate research on a particular topic [40–43]. These reviews differ from traditional systematic literature reviews in that they consider both published research and "grey" literature, such as white papers, videos, and online blogs. MLRs are being increasingly utilized in the fields of computer science and software engineering to identify credibility factors in digital news stories [44–46]. The process of conducting an MLR follows guidelines proposed by Abrar et al. [47] and is depicted in Figure 1. By including both formal and informal sources, MLRs provide a more comprehensive understanding of a topic compared to traditional systematic literature reviews, which only consider formally published research. The process of conducting a Multivocal Literature Review (MLR) is similar to that of a typical systematic mapping (SM) or systematic literature review (SLR), with the primary distinction being the inclusion of grey literature and its consideration. The process begins with the planning and design phase, in which the research goal and questions are defined. This is followed by the conduct phase, which includes several steps such as developing a search strategy and selecting sources, creating a systematic map (classification scheme), and conducting the systematic mapping, synthesis, and review. These steps, including the use of search terms, libraries, and quality assessment, are further described in our protocol paper.

For the quality assessment of grey literature, we used the SADACO approach [47] shown in Figure 2.

We adopted this methodology and found 14 credibility factors from white and grey literature. The aim of this study was to investigate the current state-of-the-art credibility factors, using a MLR approach. This method was chosen because a significant amount of information related to the research topic is available in both the formal and grey literature, including technical reports, blogs, and standards that are not typically published in academic sources. Our MLR process was established by following the guidelines put forth by Abrar et al. [47], illustrated in Figures 1 and 2. Although it closely resembles the conventional SM and SLR processes, it sets itself apart by accommodating and managing grey data. The MLR process begins with the planning and design phase, wherein the research objectives and inquiries are formulated. Afterwards, the MLR is conducted through a series of steps, such as defining the search strategy and selecting the sources, creating a systematic map (classification scheme), performing systematic mapping, synthesis, and review. Each of these steps will be explained in greater detail in the ensuing sections.

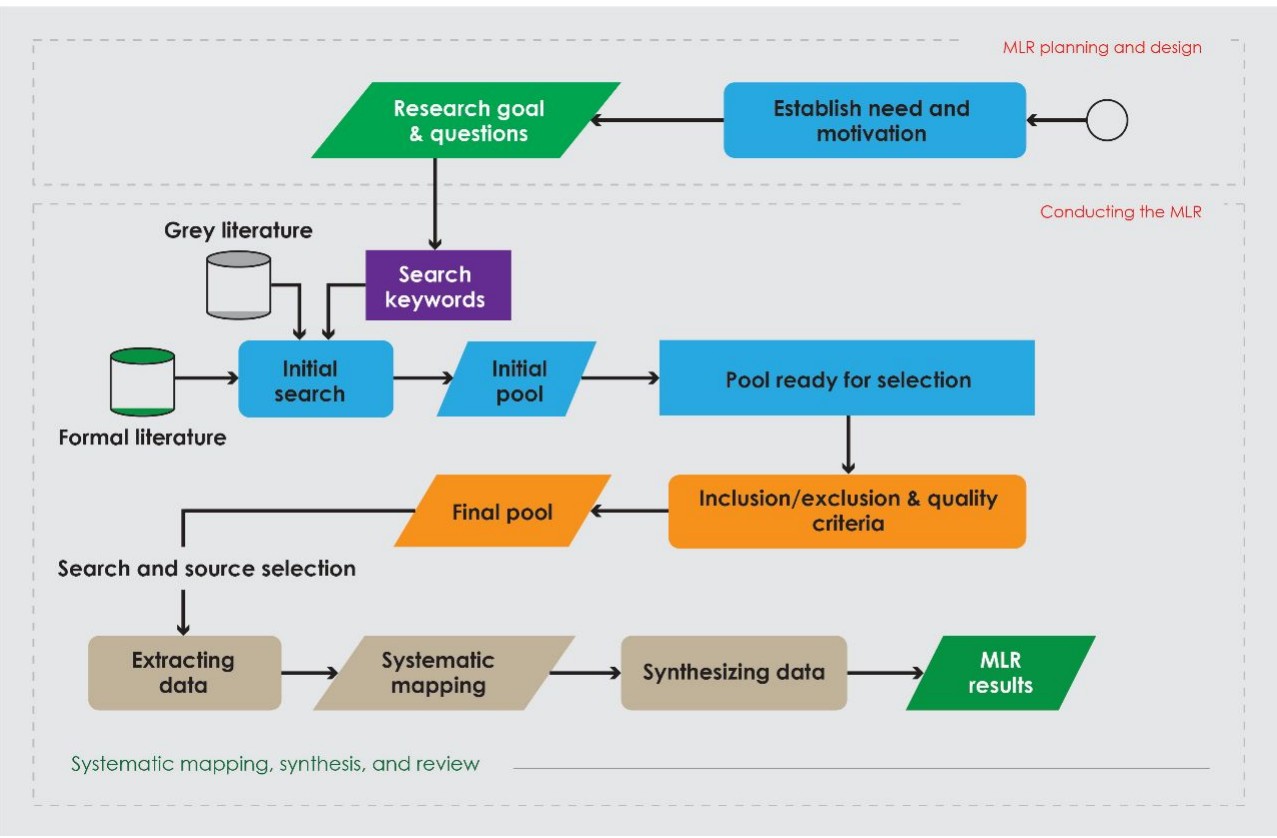

**Figure 1.** Multivocal literature review process.

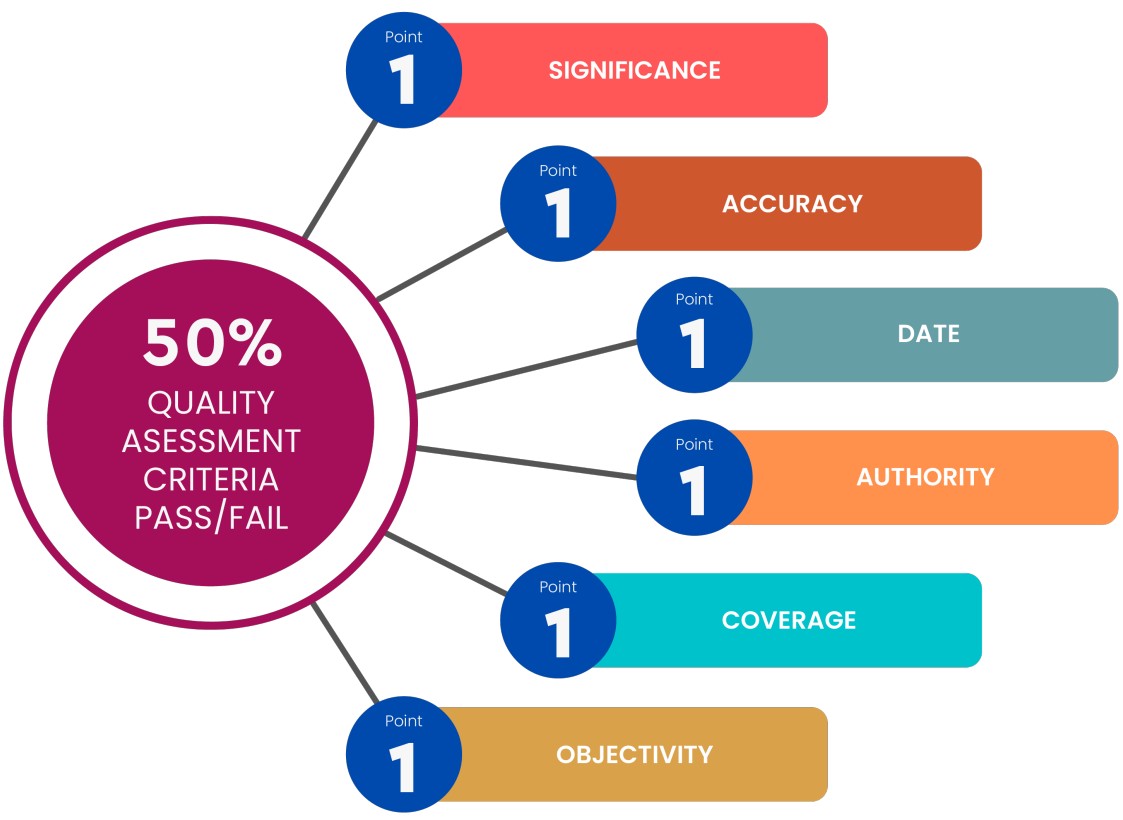

**Figure 2.** Grey literature quality assessment process SADACO [47].

*4.1. Research Goal and Questions*

The primary objectives of this research are

1. To find out the credibility factors thorough MLR.
2. To rank the identified credibility factors using the Analytical Hierarchical Process.

Our study is based on the following research questions that have motivated the work reported in this document:

**RQ 1:** What factors are identified in the literature (as well as in the grey literature) to be considered for ensuring the digital information credibility of digital news stories?

**RQ 2:** How should we rank the identified credibility factors using the Multi-Criteria-Decision-Making (MCDM) algorithm?

*4.2. Search Strategy*

The initial phase of performing an MLR involves the retrieval and curation of appropriate sources for appraisal. This necessitates devising a search strategy, explicated in this section, as well as specifying the inclusion and exclusion criteria and the benchmarks for assessing quality, described in subsequent sections. The search strategy delineates the process of locating pertinent sources and encompasses various stages. Initially, we deduced the search terms by scrutinizing the research questions' keywords from four angles: population, intervention, relevant outcome, and experimental design, as stated below:

**Population:** digital news stories.

**Intervention:** credibility factors, characteristics, practices.

**Outcomes of relevance:** to determine the factors in ensuring the credibility in digital news stories.

**Experimental Design:** multivocal literature review.

After confirming the suitability of the search terms we generated, we checked their relevance to the topic by conducting searches on both the Google search engine and academic databases, and also identified synonyms that were related. By applying Boolean operators, we combined these synonyms, and then proceeded to create various search combinations using the keywords. Eventually, we arrived at the final search string that we used in our study as shown in Table 1.

The search string was applied to the following digital libraries:

- IEEE XPLORE;
- SCIENCE DIRECT;
- ACM;
- GOOGLE SCHOLAR (SEARCH ENGINE);
- SPRINGER LINK;

The outcome of the above-mentioned research question is further analysed through some statistical and mathematical tests.

**Table 1.** Search strings and databases.

| S.NO | Database | Search String |
|---|---|---|
| 1 | GOOGLE SCHOLAR | ("Information Credibility" OR "information believability" OR "news credibility") AND ("News Stories" OR "Digital News archives") AND ("credibility factors" OR "credibility indicators" OR "Practices" OR "Solutions") |
| 2 | ACM | [[All: "information credibility"] OR [All: "digital information believability"] OR [All: "information believability"] OR [All: "news credibility"] OR [All: "news believability"]] AND [[All: "digital news stories"] OR [All: "news stories"] OR [All: "digital news archives"]] AND [[All: "credibility factors"] OR [All: "credibility indicators"] OR [All: "success factors"] OR [All: "practices"] OR [All: "solutions"]] |
| 3 | IEEE XPLORE | ("All Metadata":"Information Credibility") OR ("All Metadata":"information believability") OR ("All Metadata":"news credibility") AND ("All Metadata":"News Stories") OR ("All Metadata":"News archives") AND ("All Metadata":"credibility factors") OR ("All Metadata":"credibility indicators") OR ("All Metadata": "information credibility Practices") |
| 4 | SCIENCE DIRECT | ("Information Credibility" OR "news credibility" OR "news believability") AND ("News Stories" OR "Digital News archives") AND ("credibility factors" OR "credibility indicators" OR "Practices" OR "Solutions") |
| 5 | SPRINGER LINK | ("Information Credibility" OR "information believability" OR " news credibility") AND ("credibility factors" OR "credibility indicators" OR "Practices" OR "Solutions") |

### 4.3. Statistical Tests

The statistical methodologies employed in this study aimed to analyze the significance and correlations of credibility factors with study strategies. Techniques such as Chi-Square tests and ANOVA were utilized to provide statistical evidence and support the findings concerning the relationship between credibility factors and study strategies. To assess the significance of the identified credibility factors shown in Table 2, a Chi-Square analysis was conducted, considering variables such as Methodologies, Source libraries, and Time. The purpose of the Chi-Square test was to determine if there were any statistically significant variations among the different study strategies. The hypotheses examined aimed to establish whether there existed a notable difference among the study strategies employed for a specific credibility factor. The outcomes of the Chi-Square analysis are displayed in Table 3, providing details such as the Pearson Chi-Square value, degrees of freedom (df), and asymptotic significance. The *p*-value approach was employed to evaluate the significance of the hypotheses. By comparing the *p*-value with the predetermined significance level, decisions regarding the acceptance or rejection of the null hypothesis in favor of the alternative hypothesis were made. If the I-value was found to be lower than the significance level, the null hypothesis was rejected. To analyze the occurrences and factors, an Analysis of Variance (ANOVA) test was conducted on the dataset. This statistical test allowed for the assessment of the significance of the variables by comparing the means and variances between different groups. The appropriate null and alternative hypotheses were defined, and the test statistics for the F-statistic were calculated. The ANOVA analysis results, including the Type III Sum of Squares, degrees of freedom, Mean Square, F-value, and significance level, are presented in Table 4. These findings and analysis are discussed in Section 5.

**Table 2.** Credibility factors and their frequency distribution.

| No | CREDIBILITY FACTORS | FREQUENCY | %AGE |
|---|---|---|---|
| 1 | Number of Views | 180 | 81 |
| 2 | Reputation of the content creator | 125 | 56 |
| 3 | Content creator Followers on social media | 188 | 84 |
| 4 | Impartiality | 111 | 50 |
| 5 | Frequent Sharing | 205 | 92 |
| 6 | Number of Likes | 166 | 75 |
| 7 | Publisher's Reputations | 200 | 90 |
| 8 | Source | 212 | 95 |
| 9 | Relevancy of the contents | 163 | 73 |
| 10 | Reader's feedback | 75 | 33 |
| 11 | Background Knowledge | 145 | 65 |
| 12 | content creator's association | 219 | 99 |
| 13 | belonging to the news place | 199 | 90 |
| 14 | Latest updates | 105 | 47 |

**Table 3.** Significant credibility factors based on occurrences.

| S.NO | CREDIBILITY FACTORS | %AGE |
|---|---|---|
| 1. | Number of Views | 81 |
| 2. | Content creator Followers on social media | 84 |
| 3. | Frequent Sharing | 92 |
| 4. | Publisher's Reputations | 90 |
| 5. | Source | 95 |
| 6. | Relevancy of the contents | 73 |
| 7. | Content Creator's Association | 99 |
| 8. | Belonging to the news place | 90 |

**Table 4.** Statistical analysis results of credibility factors.

| Test | Chi-Square Value | Degrees of Freedom (df) | Significance Level |
|---|---|---|---|
| Pearson Chi-Square | 130.414 [a] | 90 | 0.003 |
| Likelihood Ratio | 76.843 | 90 | 0.837 |
| Linear-by-Linear Association | 0.198 | 1 | 0.657 |
| N of Valid Cases | 2703 | | |

[a] one cell (0.9% ) has an expected count of less than 5. The minimum expected count is 3.84.

### 4.4. Applying Analytical Hierarchical Process (AHP)

In the second phase of the study, the analytical hierarchy process (AHP) was applied to rank the listed credibility factors and their respective categories. AHP is a multi-criteria decision-making technique developed by Saaty [48] and is known for being a precise and accurate method for ranking and prioritizing items. It has also been used in other research areas to solve complex decision-making problems [49]. The steps involved in implementing AHP are shown in Figure 3. One of the advantages of using AHP in this study is that it is well-suited for analyzing data obtained through the survey process. The steps depicted in Figure 3 are explained in the following sections.

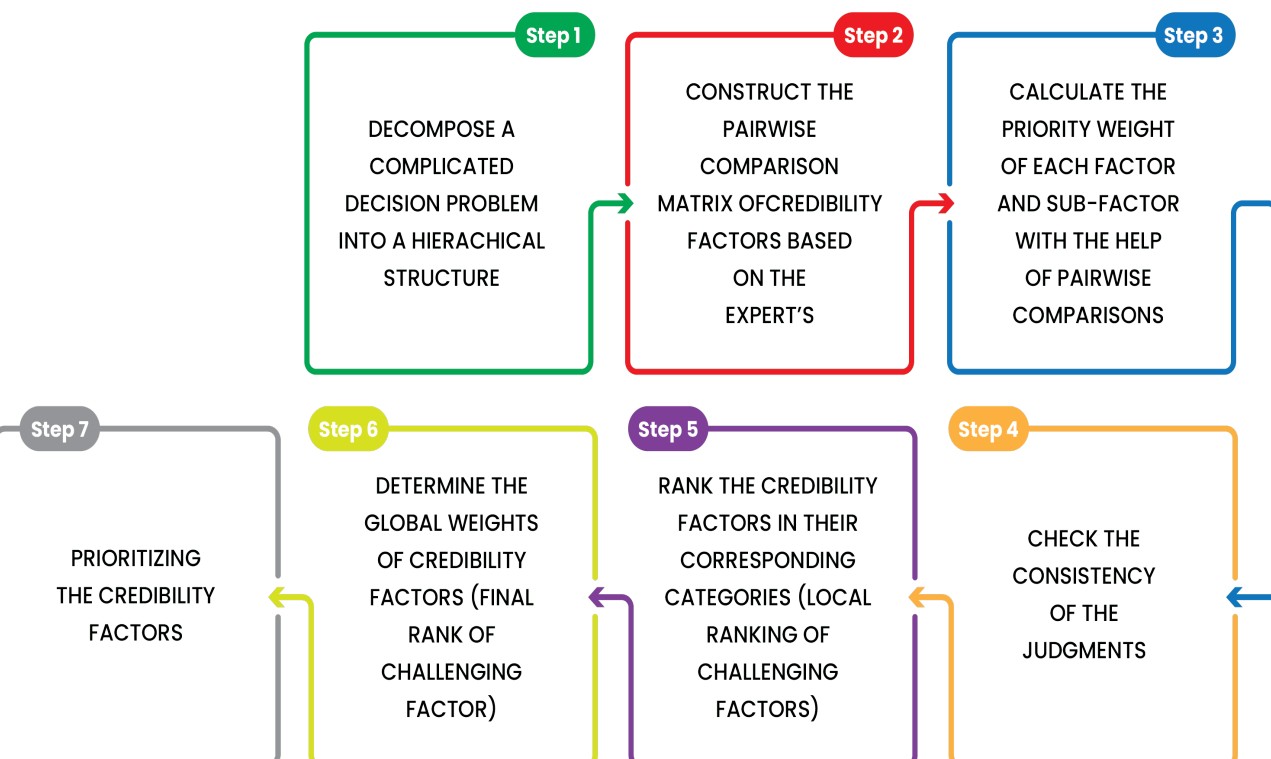

**Figure 3.** Analytical Hierarchical Process (AHP) stages.

## 5. Results

During the data extraction phase, we carefully selected a range of literature sources that were relevant to our research goals. These sources included both grey literature, such as reports and surveys, and white literature, such as journal articles and academic papers. We used a pre-defined data extraction form to carefully extract the necessary data from these sources in a standardized and organized manner. The data extraction form was designed to capture a variety of important details about each literature source. This included the date of review, the title of the publication, and the names of the authors. We also recorded the reference for each source and the database from which it was obtained. Additionally, we recorded various credibility factors that have a positive impact on the trustworthiness of the information, such as the methodology used in the study and the publication quality. We also recorded information about the target population and sample population of each study, as well as the type of organization conducting the analysis. This could include media cells, news agencies, or research institutes. We noted the size of the company, as well as the country or location of the analysis. We recorded the year of publication and the type of news covered, such as political, sports, or cultural topics. Finally, we documented whether the news was domestic or foreign in nature and the medium through which it was disseminated, such as television, blogs, or social media. The data were extracted on the pre-defined extraction form from each of the finally selected sources. Our final selection

includes a sample size of 161 papers from grey literature and 61 from white literature. After the data extraction phase, the data synthesis was performed to identify the credibility factors from the extracted data. The data synthesis phase was conducted by the primary reviewer (the primary author) with the help of a secondary reviewer (the co-author). After a thorough review with the Research Evaluation team, we have identified 14 credibility factors from all sources. Table 2 contains list of the credibility factors found from the literature adopting a multivocal literature review.

The debate on some of the credibility factors are discussed below;

**Sources** : The "source" of a digital news story refers to the origin of the information contained in the story [50]. It can refer to the person or organization that provided the information, as well as the media outlet or platform that published the story. The credibility of the source is a crucial factor in assessing the credibility of a digital news story because it determines the reliability and accuracy of the information presented [51]. If the source is credible, meaning that it has a proven track record of providing accurate and unbiased information, the news story is more likely to be credible. On the other hand, if the source is unreliable or has a history of publishing inaccurate or biased information, the news story may be less credible or even misleading [52]. In the digital age, where anyone can publish news or information online, it is important to be critical of sources and to verify information before accepting it as true. This is especially important in the context of social media, where news stories can spread quickly and without any fact-checking or editorial oversight. By paying attention to the source of a digital news story, readers can better assess its credibility and make informed decisions about whether or not to believe it [50–52].

**Number of Views**: The number of views that a digital news story receives can impact its perceived credibility in a number of ways [53]. On the one hand, a high number of views can suggest that the story is important or newsworthy, and that many people have found it to be credible and worth sharing with others. This can give the story a certain level of legitimacy and authority, as it has been deemed worthy of attention by a large number of people [54].

**Frequent Sharing**: Frequent sharing of digital news stories has a positive effect on increasing their reach and visibility, but can also lead to the spread of misinformation and a lack of in-depth understanding [55]. On one hand, the frequent sharing of digital news stories can have a positive effect on their reach and visibility. With the rise of social media and instant messaging apps, news stories can be shared quickly and easily, increasing their potential audience. This can be especially beneficial for stories that may not have received as much attention otherwise. Frequent sharing can also increase the likelihood of a story going viral, resulting in even greater exposure [55–58]. On the other hand, frequent sharing can also have negative consequences. One major issue is the spread of misinformation, as stories are shared without being fact-checked or vetted for accuracy [53]. This can be especially problematic in today's world, where fake news and conspiracy theories can spread quickly and easily online. Additionally, frequent sharing may encourage a "clickbait" mentality, where news outlets prioritize sensational headlines over in-depth reporting and analysis [54,55]. Another issue with frequent sharing is the lack of in-depth understanding that can result. With so many stories being shared on a daily basis, it can be difficult for readers to fully comprehend the context and significance of each individual story. This can lead to a "surface-level" understanding of current events, without fully grasping the nuances or complexities of the issues at hand [54–56].

*5.1. Analysis by Applying Statistical Tests*

After identifying the credibility factors in digital news stories through MLR, we classified a few factors in Table 3 based on their significance. The criteria for selection of a significant credibility factor as a credibility factor will be considered as a significant credibility factor whose frequency was ≥70 [16]. The identified considerable credibility factors are "Number of Views", "Content creator Followers on social media", "Frequent Sharing", "Number of Likes", "Publisher's Reputations", "Source", "Relevancy of the

contents", "Content Creator's Association" and "Belonging to the news place." These credibility factors were further evaluated for significance. Hence, we applied Chi-Square statistical analysis based on varinews like methodologies, source libraries, and time. The aim is to recognize whether these credibility factors remain stable/consistent in each methodology, source library, and time, respectively, or vice versa. We used the Chi-Square test to identify statistically significant differences among the various study strategies. The test statistic for the Chi-square is given by the following expression as

$$\chi^2 = \sum_{i=1}^{r} \sum_{j=1}^{c} \frac{\left(o_{ij} - e_{ij}\right)^2}{e_{ij}}$$

In the context of a contingency table, $o_{ij}$ refers to the observed frequency, representing the actual count in the ith row and jth column. Conversely, $e_{ij}$ represents the expected frequency, which is the anticipated count in the *i*th row and *j*th column assuming that the row and column variables are independent (as stated by the null hypothesis). Additionally, '*r*' signifies the total number of rows present in the contingency table, while '*c*' signifies the total number of columns. To evaluate the significance of the credibility factors, we conducted a Chi-Square statistical analysis based on variables such as methodologies, source libraries, and time. The aim was to determine if these credibility factors remain consistent within each methodology, source library, and time, or vice versa. The Chi-Square test was employed to identify statistically significant differences among the study strategies. Null and alternative hypotheses were examined for the credibility factors as follows:

**H0.** *There is no significant difference among the various study strategies used for a particular credibility factor.*

**H1.** *There is a significant difference among the various study strategies used for a particular motivator.*

A significance level of 0.05 (or 5%) was employed in the analysis, as *p*-values below this threshold are generally regarded as statistically significant. We have used the "SPSS" software for the analysis of these data. The results are presented in Table 4.

**Concept of *p*-Value approach:** *p*-value is the minimum significance level value based on which we want to reject Ho. If the *p*-value is greater than the significance level, the null hypothesis cannot be rejected, while the alternative hypothesis is accepted when the *p*-value is less than the significance level. The results of our data are exhibited in Table 4.

The chi-square test was performed on the credibility factors of the variable methodologies, i.e., Systematic Literature Review, Ordinary Literature Review, Case Study, Survey, and Interview. The results shown in Table 4 illustrate that the *p*-value is less than the significance level. Hence, the null hypothesis is rejected, and it concluded that there is a significant association between the factors and the study strategies. Further, we performed the ANOVA test on our data set. The result is illustrated in Table 5. The appropriate null and alternative hypotheses are as follows:

**H$_0'$.** *The difference between the column mean is zero, i.e., the occurrences are all equal.*

**H$_1'$.** *Not all column means are equal.*

**H$_0''$.** *The row means are equal, or the factors are equal.*

**H$_1''$.** *Not all the factors have equal means.*

**H$_0'''$.** *There is no interaction between the column and rows.*

**H$_1'''$.** *The interaction effect is not equal to zero.*

The test statistic in this case will be used as: F = Estimated variance from "Between SS"/Estimated variance from "Error SS"

$$F = \frac{s_1^2}{s_3^2} \text{ and } F = \frac{s_2^2}{s_3^2}$$

where $S_1^2$ and $S_2^2$ derived from the "between column means SS," and the "between row means SS". And $S_3^2$ is derived from errors sum of square.

**Table 5.** Analysis of Variance (ANOVA) for occurrences and factors.

| Source | Hypothesis | Type III Sum of Squares | Degrees of Freedom (df) | Mean Square | F | Sig. |
|---|---|---|---|---|---|---|
| Intercept | Differences in Means | 65,234.009 | 1 | 65,234.009 | 294.619 | 0.000 |
| Error | | 3321.277 | 15 | 221.418 [a] | | |
| Columns | Differences in Means | 20,324.179 | 6 | 3387.363 | 124.254 | 0.000 |
| Error | | 2453.536 | 90 | 27.262 [b] | | |
| Factors | Differences in Means | 3321.277 | 15 | 221.418 | 8.122 | 0.000 |
| Error | | 2453.536 | 90 | 27.262 [b] | | |
| Columns * Factors | Differences in Means | 2453.536 | 90 | 27.262 | | |
| Error | | 0.000 | 0 | . [c] | | |

(Note': [a] MS(Factors), [b] MS (Columns * Factors), [c] MS(Error)).

The *p*-value shown in Table 5 illustrates that the computed values of the F-statistic fall in the critical regions. Hence, we reject all three null hypotheses and accept the three alternative hypotheses. The term "Type III" in the ANOVA table refers to the method used to calculate the sum of squares. It takes into account the presence of other factors in the model when assessing the variation attributed to each source. This allows for a more accurate evaluation of the individual contributions of the factors.

### 5.2. Analysis by Applying Analytical Hierarchical Process (AHP)

Analytical hierarchical processing (AHP) is a well-known method used for Multi-Criteria Decisions implemented by Satty [48]. The AHP, or Analytic Hierarchy Process, is a method used to help solve complex decision-making problems that have both quantitative and qualitative aspects. It has been widely studied and applied by researchers in a variety of fields [59–62]. Furthermore, Analytical Hierarchical Process is used to prioritize the credibility factors based on their relative importance. The AHP has two phases: the categorization of the factors and applying the AHP. The second phase comprises seven steps that are elaborated on in the subsequent sections.

#### 5.2.1. The Categorization of the Factors

In Phase 1, the credibility factors are mapped to their respective categories through experts. In this study, we have applied the MLR to identify credibility factors. Through our MLR, we identified 14 credibility factors in digital news stories, as shown in Table 2. The credibility factors listed in Table 2 were further classified into three categories: "Content Creator," "Content Background," and "Media Response." Figure 4 shows the mapping of each credibility factor to its respective category according to experts' point of view. The "Content Creator" category is comprised of "Reputation of content creator", "Content creator Followers on social media," "Background Knowledge", "content creator's association"

and "belonging to the news place." The "Content Background" category consists of "Publisher's Reputations", "Source", "Relevancy of the contents," and "Latest updates". The "Media Response" category consists of "Number of views", "Frequent Sharing, "Number of Likes", "Reader's feedback" and "Impartiality". These details are depicted in Figure 4.

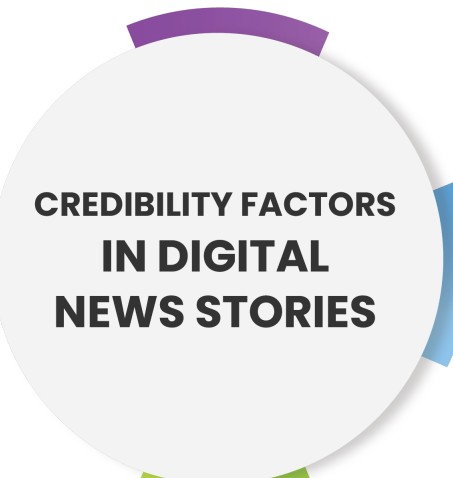

**CONTENT CREATOR**
- ▶ Reputation of content creator
- ▶ Content creator Followers on social media
- ▶ Background knowledge
- ▶ Content creator's association
- ▶ Belonging to the news place

**CONTENT BACKGROUND**
- ▶ Publisher's Reputations
- ▶ Source
- ▶ Relevancy of the contents
- ▶ Latest updates

**MEDIA RESPONSE**
- ▶ Number of views
- ▶ Frequent sharing
- ▶ Number of Likes
- ▶ Reader's feedback
- ▶ Impartiality

**Figure 4.** The hierarchical structure of credibility.

5.2.2. Decompose the Complex Problem into Its Hierarchical Structure

This step involves identifying goals and categories and ranking the credibility factors. The hierarchical structure of the problem has at least three levels, as illustrated in Figure 5. At the highest level (level 1), the goal of the problem is stated. The factors and subfactors are organized at level 2 and level 3, respectively. We created a hierarchical structure, depicted in Figure 6, based on the categorization shown in Figure 4. This structure maps the credibility factors to their relevant category according to expert opinions. The first level represents the goal of the study, the second level represents the categories of credibility factors, and the third level represents the credibility factors (subfactors).

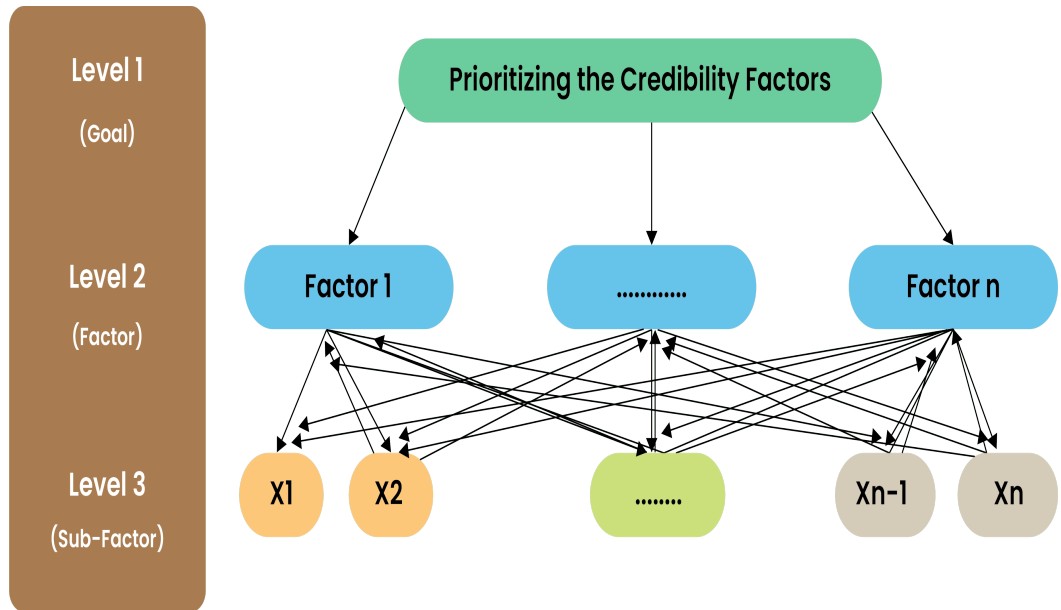

**Figure 5.** The hierarchical structure of the decision problem.

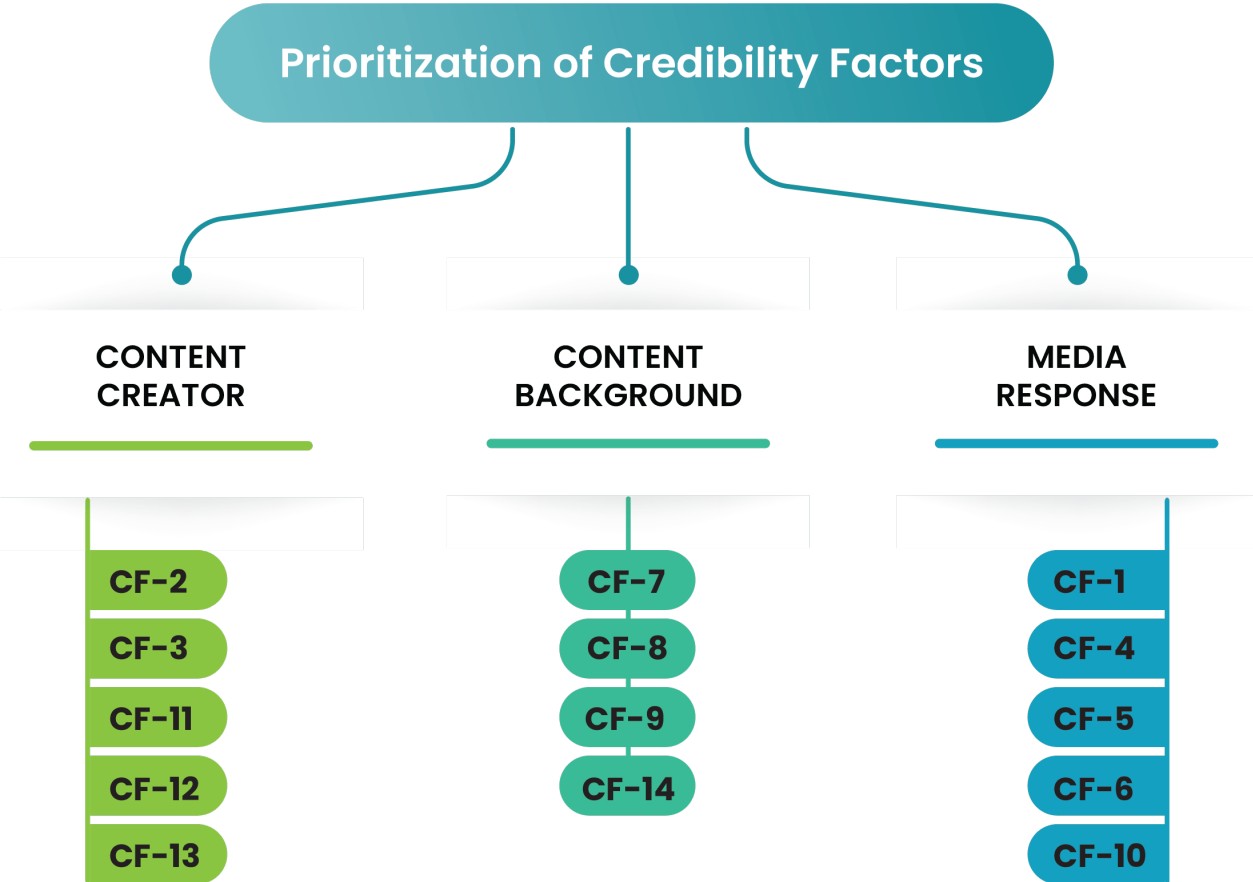

**Figure 6.** Categorization and mapping of credibility factors.

### 5.2.3. Construct a Pair-Wise Matrix of Sub-Factors to Find the Priority Weight/Vector of Credibility Factors

To apply the analytical hierarchy process (AHP) to prioritize the credibility factors and their categories, a pair-wise comparison survey was conducted. The survey was completed by 43 participants, which may raise concerns about the representatives of the sample and

the potential impact on the study's findings and conclusions. However, AHP is a subjective method that is able to handle small samples of data [6], and it has been used in other studies with relatively small sample sizes. For example, studies by Shameem et al. [49] and Cheng and Li [49] used small sample sizes of only five and nine participants, respectively, to evaluate perceptions and experiences or prioritize factors. Wong and Li [63] also used a sample of nine experts to survey intelligent building systems using the AHP method. Based on these examples, the sample size of 43 responses used in this study appears sufficient for evaluating the data collected through the AHP process. The pair-wise comparison matrix can be explained by comparing two credibility factors, i.e., CF-1 and CF-2, regarding their importance in ensuring credibility in digital news stories. CF-1 is 7 concerning CF-2 because CF-1 has a seven-degree more excellent value than CF-2, as shown in Table 6.

**Table 6.** Example of pair-wise comparison matrix.

|  | **CF-1** | **CF-2** |
| --- | --- | --- |
| CF-1 | 1 | 7 |
| CF-2 | 1/7 | 1 |

To determine the relative importance of the credibility factors and their categories, we used the same method to create pair-wise comparison matrices for each. We employed a nine-point standardized comparison scale, as shown in Table 7 and described in Table 8, to assess the significance of each factor and category.

**Table 7.** Details of the intensity scale.

| **Description** | **Significance Intensity** |
| --- | --- |
| Equally important | 1 |
| Moderately important | 3 |
| Strongly more important | 5 |
| Very strongly more important | 7 |
| Extremely more important | 9 |
| Intermediate values | 2,4,6,8 |

**Table 8.** Description of the 9-point scale for the intensity of importance.

| **Size of Matrix** | **1** | **2** | **3** | **4** | **5** | **6** | **7** | **8** | **9** | **10** |
| --- | --- | --- | --- | --- | --- | --- | --- | --- | --- | --- |
| RI | 0 | 0 | 0.58 | 0.9 | 1.12 | 1.32 | 1.41 | 1.45 | 1.45 | 1.49 |

As seen in Table 8, RI represents a Random Index, and its value changes depending on the matrix size. After collecting the survey responses, we developed the pair-wise matrix. Tables 9–12 represent the pair-wise matrix for each category. We used CF in each table, which stands for "Credibility Factor."

**Table 9.** Pair -wise matrix for the category of "Content Creator".

|  | CF2 | CF3 | CF11 | CF12 | CF13 |
|---|---|---|---|---|---|
| CF2 | 1 | 3 | 1/3 | 1/3 | 1/5 |
| CF3 | 1/3 | 1 | 1/5 | 1/5 | 1/7 |
| CF11 | 3 | 5 | 1 | 3 | 1/3 |
| CF12 | 3 | 5 | 1/3 | 1 | 1/3 |
| CF13 | 5 | 7 | 3 | 3 | 1 |

**Table 10.** Pair-wise matrix for category of "Content Background".

|  | CF7 | CF8 | CF9 | CF14 |
|---|---|---|---|---|
| CF7 | 1 | 1/7 | 1/3 | 1/7 |
| CF8 | 7 | 1 | 7 | 3 |
| CF9 | 3 | 1/7 | 1 | 1/3 |
| CF14 | 7 | 1/3 | 3 | 1 |

**Table 11.** Pair-wise matrix for category of "Media Response".

|  | CF1 | CF4 | CF5 | CF6 | CF10 |
|---|---|---|---|---|---|
| CF1 | 1 | 1/3 | 1/2 | 1/5 | 1/7 |
| CF4 | 3 | 1 | 1/3 | 1/7 | 1/5 |
| CF5 | 2 | 3 | 1 | 1/2 | 1/3 |
| CF6 | 5 | 7 | 2 | 1 | 1/2 |
| CF10 | 7 | 5 | 3 | 2 | 1 |

**Table 12.** Pair -wise matrix for all categories.

|  | CONTENT CREATOR | CONTENT BACKGROUND | MEDIA RESPONSE |
|---|---|---|---|
| CONTENT CREATOR | 1 | 3 | 5 |
| CONTENT BACKGROUND | 1/3 | 1 | 3 |
| MEDIA RESPONSE | 1/5 | 1/3 | 1 |

### 5.2.4. Calculate the Priority Weight of the Credibility Factor

The pair-wise comparison is performed to calculate the factors' priority weights [64]. Based on their relative significance and the criteria specified at the higher level, credibility factors are compared at every level. The pair-wise comparison matrices are used to calculate the priority weight as follows:

1.  Matrix: pair-wise comparison matrix of the credibility factors. These pair-wise matrices are discussed in the section "construction of pair-wise matrix".
2.  Normalizing the matrix: divide each value in each column by the sum of that column. These pair-wise matrices given in section "construction of pair-wise matrix" as Tables 9–12 are then passed to another phase where we divide each value by the sum of that column. This process produces the normalized matrices.
3.  Priority weight: calculate an average of each row of a matrix for normalization.

$\lambda$ Max calculation for the Category "**Content Creator**" is given in Table 13:

$$\lambda Max = (12.3333333 * 0.087241046)+$$
$$(21 * 0.042676137) + (4.86666667 * 0.25018496)+$$
$$(7.53333333 * 0.169690354) + (2.009524 * 0.450207503)$$
$$= 5.37277528 \tag{1}$$

$$CI = (Equation\ (1) - 5)$$
$$/(5 - 1) = (5.37277528 - 5)/(5 - 1)$$
$$= 0.09319382 \tag{2}$$

$$CR = Equation\ (2)/1.12 = 0.09319382/1.12$$
$$= 0.08320877 \leq 0.1 (Consistency\ Okay). \tag{3}$$

**Table 13.** Normalized matrix for category of "Content Creator".

|      | CF2   | CF3   | CF11  | CF12  | CF13  | Priority Weight |
|------|-------|-------|-------|-------|-------|-----------------|
| CF2  | 0.081 | 0.143 | 0.068 | 0.044 | 0.1   | 0.087241046     |
| CF3  | 0.027 | 0.048 | 0.041 | 0.027 | 0.071 | 0.042676137     |
| CF11 | 0.243 | 0.238 | 0.205 | 0.398 | 0.166 | 0.25018496      |
| CF12 | 0.243 | 0.238 | 0.068 | 0.133 | 0.166 | 0.169690354     |
| CF13 | 0.405 | 0.333 | 0.616 | 0.398 | 0.498 | 0.450207503     |

The $\lambda$ Max calculation for the category "**Content Background**" is given in Table 14:

$$\lambda Max = (18 * 0.051279377) + (1.619047619 * 0.573598943)$$
$$+(11.33333333 * 0.104401335) + (4.476190476 * 0.270720345)$$
$$= 4.246723749 \tag{4}$$

$$CI = (Equation\ (4) - 4)/(4 - 1)$$
$$= (4.246723749 - 4)/(4 - 1) = 0.08224125 \tag{5}$$

$$CR = Equation\ (5)/0.9 = (0.08224125)/0.9 = 0.091379166 \leq 0.1$$
$$(Consistency\ Okay). \tag{6}$$

**Table 14.** Normalized matrix for the category of "Content Background".

|      | CF7   | CF8   | CF9   | CF14  | Priority Weight |
|------|-------|-------|-------|-------|-----------------|
| CF7  | 0.056 | 0.088 | 0.029 | 0.032 | 0.051279377     |
| CF8  | 0.389 | 0.618 | 0.618 | 0.67  | 0.573598943     |
| CF9  | 0.167 | 0.088 | 0.088 | 0.074 | 0.104401335     |
| CF14 | 0.389 | 0.206 | 0.265 | 0.223 | 0.270720345     |

The $\lambda$ Max calculation for the category **"Media Response"** is given in Table 15:

$$\begin{aligned}\lambda Max = & (18 * 0.053364915) + (16.33333 * 0.081) + \\ & (6.833333333 * 0.144882087) + (3.842857143 * 0.297802896) + \\ & (2.176190476 * 0.422800085) \\ & = 5.340553846\end{aligned} \tag{7}$$

$$\begin{aligned}CI = & (Equation\ (7) - 5)/(5 - 1) = (5.340553846 - 5)/(5 - 1) \\ & = 0.085138461\end{aligned} \tag{8}$$

$$\begin{aligned}CR = & Equation\ (8)/1.12 = (0.085138461)/1.12 \\ & = 0.076016483 \leq 0.1 (Consistency\ Okay).\end{aligned} \tag{9}$$

**Table 15.** Normalized matrix for the category of "Media Response".

|      | CF1   | CF4   | CF5   | CF6   | CF10  | Priority Weight |
|------|-------|-------|-------|-------|-------|-----------------|
| CF1  | 0.056 | 0.02  | 0.073 | 0.052 | 0.066 | 0.053364915     |
| CF4  | 0.167 | 0.061 | 0.049 | 0.037 | 0.092 | 0.081           |
| CF5  | 0.111 | 0.184 | 0.146 | 0.13  | 0.153 | 0.144882087     |
| CF6  | 0.278 | 0.429 | 0.293 | 0.26  | 0.23  | 0.297802896     |
| CF10 | 0.389 | 0.306 | 0.439 | 0.52  | 0.46  | 0.422800085     |

A $\lambda$ Max calculation for **all categories** is given in Table 16:

$$\begin{aligned}\lambda Max = & (1.533 * 0.6333) + (4.333 * 0.2605) + (9.000 * 0.1062) \\ & = 3.055361493\end{aligned} \tag{10}$$

$$\begin{aligned}CI = & (Equation\ (10) - 3)/(3 - 1) = (3.055361493 - 3)/(3 - 1) \\ & = 0.027680747\end{aligned} \tag{11}$$

$$\begin{aligned}CR = & Equation\ (11)/0.58 = 0.027680747/0.58 \\ & = 0.047725425 \leq 0.1 (Consistency\ Okay).\end{aligned} \tag{12}$$

**Table 16.** Normalized matrix for overall categories.

|                      | CONTENT CREATOR | CONTENT BACK-GROUND | MEDIA RESPONSE | PRIORITY |
|----------------------|-----------------|---------------------|----------------|----------|
| CONTENT CREATOR      | 0.6522          | 0.6923              | 0.5556         | 0.6333   |
| CONTENT BACKGROUND   | 0.2174          | 0.2308              | 0.3333         | 0.2605   |
| MEDIA RESPONSE       | 0.1304          | 0.0769              | 0.1111         | 0.1062   |

5.2.5. Perform Consistency Check

Priority factors are only acceptable when the Consistency Ratio value is less than 0.1, and Consistency Ratio (CR) values up to 0.1 are acceptable. To enhance the consistency of the pair-wise table, the procedure should be repeated if the CR values are not within the recommended range. To calculate the pair-wise matrix consistency, a consistency ratio

(CR) and consistency index (CI) are used in AHP [49,63]. Using the following equation, we can check the consistency of the pair-wise matrix : CI = $\frac{\lambda\ max - n}{n-1}$ Where CI represents the consistency index, $\lambda$ max is the eigenvalue of the matrix, and n represents the size of the matrix or the number of credibility factors in the matrix. We will find the consistency ratio by the below equation after finding the CI: CR = $CI/RI$ Where CR represents the consistency ratio, CI is used for the consistency index, and RI is the random consistency index illustrated in Table 7, which has constant values. Each credibility factor's weighted value (W) is calculated by averaging the normalized values of the corresponding row shown in Tables 13–16. Hence, the $\lambda$ Max calculation for each category is given above with the normalized table.

### 5.2.6. Calculate the Credibility Factor's Local Weight (LW) and Global Weight (GW)

The local weight of a credibility factor is the priority weight assigned to each credibility factor inside its respective category. As a result, all the credibility factors' priority weights relative to their categories are calculated and listed in this stage. The value of the local weight inside each category multiplied by the value of the local weight of the corresponding category produces the value of the global weight of each Credibility Factor. We have calculated both LW and GW, which are displayed in Table 17.

**Table 17.** Summary of local and global weights of credibility factors and their rankings.

| Categories | Categories Weight | Factors | Local Weights | Local Ranking | Global Weights | Final Priority |
|---|---|---|---|---|---|---|
| CONTENT CREATOR | 0.6333 | CF2 | 0.0872 | 4 | 0.0553 | 6 |
| | | CF3 | 0.0427 | 5 | 0.027 | 10 |
| | | CF11 | 0.2502 | 2 | 0.1585 | 2 |
| | | CF12 | 0.1697 | 3 | 0.1075 | 4 |
| | | CF13 | 0.4502 | 1 | 0.2851 | 1 |
| CONTENT BACKGROUND | 0.2605 | CF7 | 0.0513 | 4 | 0.0134 | 12 |
| | | CF8 | 0.5736 | 1 | 0.1494 | 3 |
| | | CF9 | 0.1044 | 3 | 0.0272 | 9 |
| | | CF14 | 0.2707 | 2 | 0.0705 | 5 |
| MEDIA RESPONSE | 0.1062 | CF1 | 0.0534 | 5 | 0.0057 | 14 |
| | | CF4 | 0.0812 | 4 | 0.0086 | 13 |
| | | CF5 | 0.1449 | 3 | 0.0154 | 11 |
| | | CF6 | 0.2978 | 2 | 0.0316 | 8 |
| | | CF10 | 0.4228 | 1 | 0.0449 | 7 |

### 5.2.7. Identify and Create the Overall Priority Ranking

The final list of credibility factors in digital news stories, based on each credibility factor's global weight, is created in this step. Credibility factors are considered highly ranked if they have a more excellent global weight value across all categories.

The study identified 14 different factors and ranked them based on their global weight, with higher global weight values indicating greater importance. The most important factor was "CF13: Belonging to the news place," while the least important factor was "CF1: Number of views." The findings are summarized in Table 18. These results suggest that the "number of views" a digital news story receives is a lesser importance of its credibility, while the news "Belonging To the News Place" suggested a high credibility. It is important to note that the specific factors and their rankings may vary depending on the specific context and research method used in the study.

**Table 18.** Prioritization of credibility factors.

| S.NO | Credibility Factors | Priority |
|---|---|---|
| CF13 | Belonging To the News Place | 1 |
| CF11 | Background Knowledge | 2 |
| CF8 | Source | 3 |
| CF12 | Content Creator's Association | 4 |
| CF14 | Latest Updates | 5 |
| CF2 | Reputation of Content Creator | 6 |
| CF10 | Reader's Feedback | 7 |
| CF6 | Number of Likes | 8 |
| CF9 | Relevancy of The Contents | 9 |
| CF3 | Content Creator Followers on social media | 10 |
| CF5 | Frequent Sharing | 11 |
| CF7 | Publisher's Reputations | 12 |
| CF4 | Impartiality | 13 |
| CF1 | Number of views | 14 |

## 6. Implementation Challenges

Undertaking a multivocal literature review presents several challenges. Firstly, the identification and selection of pertinent sources from diverse disciplines and perspectives demand careful consideration and a comprehensive understanding of the research topic. Moreover, the diversity of sources, including variations in language, culture, and epistemology, complicates the comparison and synthesis of different perspectives and voices. Integrating these divergent viewpoints, which may entail distinct assumptions, values, and conceptual frameworks, poses an additional challenge. Ensuring the quality and credibility of sources used in the review is also crucial, but can be demanding due to varying levels of reliability and trustworthiness. Finally, analysing and synthesising information obtained from diverse sources necessitates a deep understanding of the research topic and the ability to identify key themes and concepts across multiple perspectives. Implementing the Analytic Hierarchy Process (AHP) brings forth its own array of challenges. Primarily, the subjective nature of pairwise comparisons in AHP introduces the potential for biases and inconsistencies in the decision-making process. Handling the complexity of breaking down intricate decisions into smaller components and assessing their relative importance can be time-consuming and arduous, particularly when dealing with a multitude of criteria and alternatives. Ensuring the quality of data used in AHP analysis is paramount, as flawed or incomplete data can compromise the accuracy and reliability of results. Effective communication among multiple stakeholders is essential to navigate challenges related to collaboration and mitigate potential misunderstandings. Lastly, possessing the requisite technical skills and expertise is critical for the successful implementation of the AHP method, and decision-makers may require adequate training and support to fully comprehend and apply the methodology.

## 7. Conclusions

The study aimed to identify the factors that contribute to the credibility of digital news stories by analysing both white and grey literature. This research is crucial in addressing the pressing issue of fake news and misinformation, as it provides valuable insights for developing strategies to evaluate and verify the accuracy and reliability of digital news information. We adopted an MLR and identified 14 credibility factors from this literature. They then analysed the results of their research using statistical and mathematical tests.

In order to achieve the goals of this study, we have adopted the multivocal literature review. Systematic mapping (SM) and systematic literature reviews (SLRs) are research methods that are commonly used in the fields of computer science and engineering to summarize the existing knowledge and evidence on a particular topic. These methods involve systematically searching for and reviewing relevant research studies and organizing the results in a structured and systematic way. This study identified and ranked various factors that contribute to the credibility of digital news stories. The study identified 14 different factors and ranked them based on their global weight, with higher global weight values indicating greater importance. The most important factor was "CF13: Belonging to the news place," while the least important factor was "CF1: Number of views." These results suggest that the number of views a digital news story receives is a strong indicator of its credibility, while the news source's reputation is of lesser importance. It is important to note that the specific factors and their rankings may vary depending on the context and research method used. Different studies may identify different factors as being most important, and the relative importance of each factor may vary depending on the specific goals or focus of the study. Additionally, the specific ranking of the factors may be influenced by the method used to calculate the global weight values. Despite these limitations, the study provides valuable insights into the factors that contribute to the credibility of digital news stories and can inform the development of strategies for evaluating and verifying the accuracy and reliability of this information. The multivocal literature review (MLR) allowed us to gather enough evidence to support our conclusions about credibility factors in digital information. The MLR was conducted in a systematic manner and the sources were carefully evaluated using predetermined quality criteria. This thorough approach increases our confidence in the credibility of certain news sources and reduces the risk of invalid conclusions.

## 8. Limitations and Future Directions

This para focuses on the challenges to validity and how they are resolved to build trust in the study's findings. Missing sources that were published in other databases or after conducting the research represent one of the potential threats. To obtain as much information as possible, we used an MLR to compile all relevant articles that had been published in both formal and informal literature. We searched a total of five electronic databases. This guaranteed thorough coverage of all sources. We, therefore, have enough evidence to conclude that these sources address most information credibility elements. Another drawback is that the credibility factors might occasionally be subjective. Based on our own experiences and the knowledge we obtained from the MLR, we carried out the assignment. The factors were iteratively validated by a team of five researchers from various universities to reduce this threat. The credibility factors that we obtained because of our MLR are a good representation of the actual factors in information sciences. However, because some sources do not provide adequate or clear information relating to our research topics, source selection and data extraction in the MLR are subjective. To address this potential threat, careful screening and selection of sources was conducted based on a set of quality criteria. The quality criteria were designed for both white and grey literature in our previous study.

One potential direction for future research on the topic of digital news credibility could be to expand upon the factors identified in the study and to examine the relationships between these factors and the credibility of digital news stories. This could involve conducting additional research to gather more data on the factors and their impact on credibility and using these data to develop more nuanced and refined models for assessing the credibility of digital news stories. Another potential direction for future research could be to explore the specific strategies and methods that can be used to evaluate and verify the accuracy and reliability of digital news stories. This could involve examining the effectiveness of different approaches to fact-checking or developing new tools and technologies to help people assess the credibility of news articles. Overall, the future direction of research on the topic of digital news credibility will likely be influenced by the continued evolution

of the media landscape and the emergence of new challenges and opportunities in the field. By staying abreast of these developments and continuing to explore the factors that contribute to the credibility of digital news stories, it will be possible to better understand and address the challenges of ensuring the accuracy and reliability of information in the digital age.

**Author Contributions:** Conceptualization, M.F.A. and M.S.K.; methodology, M.F.A. and I.K.; software, S.A.; validation, M.F.A., M.S.K. and S.A.; formal analysis, M.F.A.; investigation, M.S.K; resources, S.A.; data curation, M.S.K.; writing—original draft preparation, M.F.A.; writing—review and editing, M.F.A., M.E. and M.S.K.; supervision, M.S.K.; project administration, M.S.K.; funding acquisition, S.A. and M.E. All authors have read and agreed to the published version of the manuscript.

**Funding:** This research was supported by the EIAS Data Science and Blockchain Lab, College of Computer and Information Sciences, Prince Sultan University, Riyadh 11586, Saudi Arabia.

**Data Availability Statement:** The data presented in this study are available on request from the authors.

**Acknowledgments:** The authors would like to acknowledge Prince Sultan University and EIAS: Data Science and Blockchain Laboratory for their valuable support.

**Conflicts of Interest:** The authors declare no conflict of interest.

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
