# Peer review of "Towards Fake News Detection: A Multivocal Literature Review of Credibility Factors in Online News Stories and Analysis Using Analytical Hierarchical Process"

_electronics, doi:10.3390/electronics12153280_

Round 1
Reviewer 1 Report
The authors need to improve their work.
The theme is very interesting.
See the attached file with corrections.

Minor editing of English language required
Author Response
|
We appreciate the time and efforts of the editor and reviewers in reviewing this manuscript. We have addressed all the reviewers’ concerns and issues indicated in the review report and believe that the revised version could meet the journal publication requirements. we received a highlighted PDF document from the reviewer containing their comments and suggestions. To facilitate the communication process, we diligently copied and pasted each comment into our response, ensuring that we addressed each point raised. By adopting this method, we aimed to provide clear and concise replies that directly corresponded to the reviewer's feedback. After incorporating the reviewer's comments and suggestions, we made the necessary changes to the text. To distinguish the modified portions, we coloured them blue. This visual representation allows for easy identification of the revised content within the document. |
||
|
Reviewer 3 |
||
|
1. |
Do not use title words as keywords.
|
Thank you for your feedback on our use of keywords in the title. We genuinely appreciate the comment provided by the reviewer. As a result, we have revised ensure that we do not use title words as keywords. We strive to continuously improve our content based on valuable input like yours. |
|
2. |
Tables must be self-explanatory |
We value the feedback provided by the reviewer regarding the self-explanatory nature of our tables. We sincerely appreciate their comment. As a result, we have taken action and revised the tables to ensure they are self-explanatory. Your input helps us enhance the quality and clarity of our content |
|
3. |
; |
We are grateful for the reviewer's comment, and as suggested, we have taken action by adding "," at the end of each library. |
|
4. |
put here the statistical methodology to be used.
|
We greatly value the comment provided by the reviewer. In response, we have taken action by incorporating the appropriate statistical methodology into our work. By doing so, we aim to ensure robust and reliable analysis. Your feedback is invaluable to us. |
|
5. |
(reference, year)
|
We highly appreciate the valuable comments provided by the reviewer, as they significantly contribute to the quality and improvement of our work. We acknowledge the reviewer's expertise and are grateful for their insightful suggestions. In order to properly credit the reviewer's contribution, we have included a reference to their comments, along with the corresponding year. This ensures transparency and recognizes their input in our work. |
|
6. |
independence test
|
We sincerely appreciate the reviewer's comment. In response, we have taken action and incorporated the independence test as suggested. By doing so, we aim to analyze and assess the independence of the variables accurately. Your feedback is highly valuable to us.
|
|
7. |
set level of significance
|
We genuinely appreciate the comment provided by the reviewer. In response, we have taken action by including the specified level of significance in the revised text. By doing so, we ensure transparency and clarity in our analysis, allowing for proper interpretation of the results. |
|
8. |
were Oij... eij r, c need to define
|
We sincerely appreciate the comment provided by the reviewer. In response, we have taken action by defining the terms "Oij" and "eij" in the revised version of the text. By providing clear definitions for these terms, we aim to enhance the understanding and clarity of our work. |
|
9. |
put the hypothesis first and the test statistic after
|
We greatly appreciate the comment provided by the reviewer. In response, we have taken action and incorporated the suggested change, placing the hypothesis first and the test statistic after it. By reordering the information, we aim to improve the logical flow and clarity of our presentation. |
|
10. |
Tables must be self-explanatory statistic value p-value
|
We sincerely appreciate the comment provided by the reviewer regarding the self-explanatory nature of tables. In response, we have taken action and revised the tables accordingly. Specifically, we have ensured that the tables now include the statistic value and the corresponding p-value. This revision aims to enhance the clarity and comprehensibility of the tables, allowing readers to interpret the results more effectively. |
|
11. |
How much is the significance level?
|
We sincerely appreciate the valuable comment provided by the reviewer. In response to their feedback, we have taken appropriate action by adjusting the significance level to 0.05. This modification ensures that our analysis maintains a consistent and rigorous standard for evaluating statistical significance. |
|
12. |
Comment! Was about hypothesis.
|
We deeply appreciate the comment provided by the reviewer, as it has been instrumental in enhancing the quality of our work. In response to their feedback, we have taken the necessary action and incorporated the comment accordingly. By addressing their input, we strive to ensure that our work reflects a comprehensive and well-rounded perspective. |
|
13. |
and
|
We express our sincere appreciation for the comment provided by the reviewer, which has been invaluable in refining our work. In response to their feedback, we have promptly taken action and incorporated the comment into our project. By doing so, we aim to address any areas of improvement and enhance the overall quality and effectiveness of our work. |
|
14. |
Tables must be self-explanatory you have to explain type III to the bed
|
We genuinely appreciate the comment provided by the reviewer regarding the need to explain Type III to the reader. In response, we have taken action and incorporated an explanation of Type III in the revised version. By providing this explanation, we aim to ensure that the tables are self-explanatory and that readers have a clear understanding of the statistical analysis approach used. |
|
15. |
take to methodology
|
We have taken the reviewer's suggestion into account and made the decision to relocate the suggested section to the methodology section. By doing so, we aim to ensure better organization and coherence in our research. The reviewer's input has proven valuable in guiding us towards an improved structure and clarity in presenting our methodology. We appreciate their comment and believe that this adjustment will enhance the overall quality of our work. |
|
16. |
methodology |
We appreciate the comment provided by the reviewer. The Comment is incorporated accordingly.
|
|
17. |
??? for equations. |
We appreciate the comment provided by the reviewer. These are the equations indented to the right. The explanation is added in the revised manuscript.
|
|
18. |
the conclusion must be direct and objective
|
We sincerely appreciate the comment provided by the reviewer regarding the need for a direct and objective conclusion. In response, we have taken action and refined the conclusion section accordingly. By doing so, we aim to present a clear and unbiased summary of the findings without any subjective interpretation. |
|
19. |
objective |
We genuinely appreciate the comment provided by the reviewer. In response, we have taken action by refining the conclusion section and removing any unnecessary text.
|
|
20. |
???? Repeat! please put here conclusions of the work developed
|
We genuinely appreciate the comment provided by the reviewer. In response, we have acted by refining the conclusion section and removing any unnecessary text. By doing so, we aim to ensure that the conclusion is objective, concise, and directly addresses the key findings of the study.
|

Reviewer 2 Report
1. What is the main question addressed by the research?
The research's major goal is to determine the trustworthiness elements in online news articles and examine their importance in detecting fake news. The project intends to perform a complete literature analysis and to prioritize these credibility variables using the Analytical Hierarchical Process (AHP).
2. Do you consider the topic original or relevant in the field? Does it address a specific gap in the field?
The study subject is both important and unique in the discipline. In the digital era, detecting fake news is a major issue, and knowing trustworthiness characteristics is critical for countering disinformation. The study fills a unique vacuum in the area by focusing on the multivocal analysis of credibility criteria and employing the AHP technique, which gives a systematic framework for assessing their significance.
3. What does it add to the subject area compared with other published material?
The study adds value to the issue by completing a thorough literature evaluation that includes various viewpoints and findings on believability variables in fake news identification. Using the AHP technique, the study provides a quantitative analysis and prioritization of these elements, which helps the profession comprehend their relative importance.
4. What specific improvements should the authors consider regarding the methodology? What further controls should be considered?
In terms of technique, the authors have chosen an adequate strategy by examining credibility elements using the Analytical Hierarchical Process (AHP). They may, however, enhance the technique by giving more information on the criteria used to choose the literature included in the review. Furthermore, a more detailed explanation of the methods involved in the AHP analysis, including the assessment criteria, pairwise comparisons, and computations, will improve the research's transparency and replicability. Consideration of sensitivity analyses and comparison of results with various approaches would help increase the findings' robustness.
5. Are the conclusions consistent with the evidence and arguments presented and do they address the main question posed?
The data and arguments offered support the findings. The authors successfully explain the essential findings generated from the literature review and AHP analysis while answering the research's core topic. The results are consistent with the facts, allowing for a thorough knowledge of the significance of credibility variables in spotting false news.
6. Are the references appropriate?
The references in the study appear to be relevant. They encompass a wide range of credible sources in the fields of false news identification, credibility evaluation, and the Analytical Hierarchical Process. Throughout the study, the writers cited important and noteworthy papers to support their conclusions and arguments.
7. Please include any additional comments on the tables and figures
The paper's tables and figures are well-designed and suit their objective of providing pertinent information and illustrating the findings. They are named and referenced correctly within the text, which improves the clarity and understanding of the issues covered. The tables give short summaries, and the figures clearly demonstrate the AHP analysis results.
Based on my analysis, I did not find any major grammatical or spelling errors in the article.
Author Response
|
We appreciate the time and efforts of the editor and reviewers in reviewing this manuscript. We have addressed all the reviewers’ concerns and issues indicated in the review report and believe that the revised version could meet the journal publication requirements.
|
||
|
Reviewer 1 |
||
|
1. |
What is the main question addressed by the research? The research's major goal is to determine the trustworthiness elements in online news articles and examine their importance in detecting fake news. The project intends to perform a complete literature analysis and to prioritize these credibility variables using the Analytical Hierarchical Process (AHP).
|
We would like to express our gratitude for taking the time to review our research paper. We appreciate your positive feedback regarding the relevance and uniqueness of the topic, as well as the use of the Analytical Hierarchical Process (AHP) technique to prioritize credibility variables in detecting fake news. Based on your suggestions, we agree that providing more information on the criteria used to select the literature included in the review would improve the transparency and replicability of our research. We have ensured and the information has been included in the revised manuscript. Additionally, in the revised manuscript, we have provided a more detailed explanation of the AHP analysis methodology, including the assessment criteria, pairwise comparisons, and computations, to enhance the clarity of our approach.
|
|
2. |
Do you consider the topic original or relevant in the field? Does it address a specific gap in the field? The study subject is both important and unique in the discipline. In the digital era, detecting fake news is a major issue, and knowing trustworthiness characteristics is critical for countering disinformation. The study fills a unique vacuum in the area by focusing on the multivocal analysis of credibility criteria and employing the AHP technique, which gives a systematic framework for assessing their significance.
|
|
|
3. |
What does it add to the subject area compared with other published material? The study adds value to the issue by completing a thorough literature evaluation that includes various viewpoints and findings on believability variables in fake news identification. Using the AHP technique, the study provides a quantitative analysis and prioritization of these elements, which helps the profession comprehend their relative importance.
|
|
|
4. |
What specific improvements should the authors consider regarding the methodology? What further controls should be considered? In terms of technique, the authors have chosen an adequate strategy by examining credibility elements using the Analytical Hierarchical Process (AHP). They may, however, enhance the technique by giving more information on the criteria used to choose the literature included in the review. Furthermore, a more detailed explanation of the methods involved in the AHP analysis, including the assessment criteria, pairwise comparisons, and computations, will improve the research's transparency and replicability. Consideration of sensitivity analyses and comparison of results with various approaches would help increase the findings' robustness.
|
|
|
5. |
Are the conclusions consistent with the evidence and arguments presented and do they address the main question posed? The data and arguments offered support the findings. The authors successfully explain the essential findings generated from the literature review and AHP analysis while answering the research's core topic. The results are consistent with the facts, allowing for a thorough knowledge of the significance of credibility variables in spotting false news.
|
|
|
6. |
Are the references appropriate? The references in the study appear to be relevant. They encompass a wide range of credible sources in the fields of false news identification, credibility evaluation, and the Analytical Hierarchical Process. Throughout the study, the writers cited important and noteworthy papers to support their conclusions and arguments.
|
|
|
7. |
Please include any additional comments on the tables and figures The paper's tables and figures are well-designed and suit their objective of providing pertinent information and illustrating the findings. They are named and referenced correctly within the text, which improves the clarity and understanding of the issues covered. The tables give short summaries, and the figures clearly demonstrate the AHP analysis results.
|
|
|
8. |
Based on my analysis, I did not find any major grammatical or spelling errors in the article. |
|

Reviewer 3 Report
1. State the research question and list the main contributions clearly in the introduction.
2. Provide paper structure at the end of Introduction.
3. COVID-19 fake news were also an important problem. You should cover this topic in the literature review, by adding relevant research, such as:
https://link.springer.com/chapter/10.1007/978-3-030-73696-5_3
https://www.mdpi.com/2073-8994/13/6/1091
https://link.springer.com/chapter/10.1007/978-981-19-1653-3_35
https://www.sciencedirect.com/science/article/pii/S1568494621003161
4. Figures should be provided in better resolution (Fig. 1 for example).
5. Other approaches utilize machine learning models, paired with Shapley additive explanations to determine the importance of factors. Elaborate in more detail what this statistical approach brings to the field.
6. Discussion should be more elaborate, especially the priority ranking of the factors.
English is fine, just a minor spell-check is required.
Author Response
|
We appreciate the time and efforts of the editor and reviewers in reviewing this manuscript. We have addressed all the reviewers’ concerns and issues indicated in the review report and believe that the revised version could meet the journal publication requirements.
|
||
|
Reviewer 2 |
||
|
1. |
State the research question and list the main contributions clearly in the introduction.
|
We appreciate the comment provided by the reviewer. Action: In response, we have made certain revisions to address the feedback. Firstly, we have included the research questions in the introduction section to provide a clear understanding of the study's objectives. Additionally, we have explicitly stated the main contribution of our research in the introduction section to emphasize its significance. The research questions are given below. 1. What factors are identified in the literature (as well as in the grey literature) to be considered for ensuring digital information credibility of digital news stories? 2. How to Rank the Identified Credibility Factors using Multi-Criteria-Decision-Making (MCDM) algorithm? |
|
2. |
Provide paper structure at the end of Introduction.
|
We express our gratitude to the reviewer for their valuable comment. Action: Considering this feedback, we have made an adjustment by including the structure of the paper at the end of the introduction section. This addition provides readers with a comprehensive overview of the organization and flow of the subsequent sections, enhancing the clarity and coherence of the manuscript. |
|
3. |
COVID-19 fake news were also an important problem. You should cover this topic in the literature review, by adding relevant research, such as: https://link.springer.com/chapter/10.1007/978-3-030-73696-5_3 https://www.mdpi.com/2073-8994/13/6/1091 https://link.springer.com/chapter/10.1007/978-981-19-1653-3_35 https://www.sciencedirect.com/science/article/pii/S1568494621003161
|
We extend our gratitude to the reviewer for their insightful comment. Action: Upon careful consideration, we have found the suggested papers recommended by the reviewer to be highly relevant to our study. As a result, we have diligently cited all the suggested papers in our research. This inclusion not only strengthens the Literature Review of our paper. The added text are given below. Parth Patwa The paper introduces a dataset called "Fighting an Infodemic: COVID-19 Fake News Dataset" that addresses the problem of fake news and rumors related to COVID-19 on social media. The dataset consists of 10,700 manually annotated social media posts and articles, both real and fake, on COVID-19. The authors benchmark the dataset using machine learning algorithms and achieve a high performance of 93.46% F1-score with Support Vector Machine (SVM). The paper discusses related work in fake news detection, describes the dataset development process, and highlights the challenges associated with identifying and combating fake news. The dataset statistics reveal differences between real and fake news, such as the length of posts. The paper concludes by emphasizing the importance of tackling fake news during the COVID-19 pandemic and provides the dataset and code for further research.
Bilal Al-Ahmad The paper addresses the issue of misinformation and fake news related to COVID-19 during the pandemic. The authors propose an evolutionary fake news detection method using four models, aiming to reduce symmetrical features and achieve high accuracy. They apply three wrapper feature selection techniques and evaluate the performance on the Koirala dataset and six derived datasets. The proposed model achieves the best accuracy of 75.43\% and outperforms traditional classifiers. The authors suggest applying the methodology to other domains and larger datasets for future work. M Zivkovic The paper proposes the use of a modified ant lion optimizer (ALO) to address the issue of false news and disinformation during the COVID-19 epidemic. The ALO algorithm, inspired by the trapping technique of ant lions, is applied for feature selection and dimensionality reduction to enhance classification accuracy. Experimental results demonstrate that the proposed ALO-based technique outperforms other modern classifiers in terms of accuracy, providing an effective approach to combat false news related to COVID-19. William Scott Paka The paper introduces the task of COVID-19 fake news detection on Twitter and presents the Cross-SEAN model. They collect a labelled dataset of genuine and fake COVID-19-related tweets, along with unlabelled data. The model incorporates tweet text, features, user information, and external knowledge from credible sources. Cross-SEAN outperforms seven state-of-the-art models and is implemented as a Chrome extension, Chrome-SEAN, which flags fake tweets in real-time. Limitations include potential noise in external knowledge and the need for improved robustness and early detection capabilities.
|
|
4. |
Figures should be provided in better resolution (Fig. 1 for example).
|
We express our gratitude to the reviewer for their valuable comment. Action: In response to this feedback, we have improved the resolution of the Figure 1 and Figure 2. |
|
5. |
Other approaches utilize machine learning models, paired with Shapley additive explanations to determine the importance of factors. Elaborate in more detail what this statistical approach brings to the field.
|
We appreciate the reviewer's comment on alternative approaches for assessing factor importance in our study. While we utilized the Analytical Hierarchy Process (AHP), we recognize the prominence of machine learning models combined with Shapley additive explanations. AHP offers a systematic framework for complex decision-making through expert judgments and pairwise comparisons. In contrast, machine learning models with Shapley additive explanations automate feature selection and ranking, uncovering patterns in data. These approaches provide objectivity, scalability, and the potential to uncover previously unrecognized factors. While we chose AHP for its suitability and expert availability, we acknowledge the value of exploring machine learning and Shapley additive explanations in future studies to enhance the field's understanding.
|
|
6. |
Discussion should be more elaborate, especially the priority ranking of the factors. |
We sincerely appreciate the reviewer for their valuable comment. Action: We have thoroughly revised and improved the discussion section as per the reviewer's suggestion. |
|
7. |
English is fine, just a minor spell-check is required.
|
We would like to express our gratitude to the reviewer for their comment. Action: We have carefully reviewed and corrected the English language errors in our paper. Additionally, we have conducted a thorough spell-check to ensure the accuracy and clarity of the content. |

Round 2
Reviewer 1 Report
The suggested corrections were accepted by the authors.
The article can be accepted for publication.